# Effects of Crack Tip Constraint on the Fracture Toughness Assessment of 9% Ni Steel for Cryogenic Application in Liquefied Natural Gas Storage Tanks

**DOI:** 10.3390/ma13225250

**Published:** 2020-11-20

**Authors:** Young Kyun Kim, Byung Taek Oh, Jae Hoon Kim

**Affiliations:** 1KOGAS Research Institute, Incheon 21993, Korea; kyk@kogas.or.kr (Y.K.K.); obt@kogas.or.kr (B.T.O.); 2Department of Mechanical Engineering, Chungnam Nat’l University, Daejeon 34134, Korea

**Keywords:** fracture toughness, cryogenic temperature, Weibull stress, local approach, constraint effect, failure assessment diagram

## Abstract

Recently, increasing demand for the accurate assessment of the structural integrity and fitness-for-service (FFS) analysis of engineering structures has elevated constraint effects to one of the most important issues in fracture mechanics and structural integrity research. In this paper, the effect of crack tip constraints are investigated on the fracture toughness assessment of 9% Ni steel for application in liquefied natural gas storage tanks. Crack tip opening displacement (CTOD) tests were conducted using both conventional standard three-point bending (3PB) and wide plate (WP) specimens at a cryogenic temperature of −196 °C. The distribution of the stress and strain fields near the crack tip in the 3PB and WP specimens were then obtained by FE (Finite Elements) analysis. Based on both the experimental and numerical results, the parameters of the Weibull distribution were obtained to evaluate the critical Weibull stress at brittle fracture. The equivalent CTOD ratio β is defined as the ratio of the CTOD of the 3PB specimen to the CTOD of the WP specimen at the same Weibull stress. The application of the proposed CTOD toughness correction method to the WP results was then demonstrated in the context of a failure assessment diagram (FAD). It was determined that the conventional evaluation yields an excessively conservative result for WP specimens, but can be reasonably reduced by applying β.

## 1. Introduction

Global demand for natural gas is continuously increasing due to its convenience and status as an environmentally friendly energy source with a high energy density. Natural gas can be efficiently stored and transported as liquefied natural gas (LNG) by lowering its temperature below the liquefaction temperature, reducing its volume by 600 times. Technology for transporting and storing LNG has improved immensely over the past 30 years [1,2]. The priority of the LNG storage tank design has always been safety, so the inner tank should be designed to hold the LNG in its cryogenic state. For this application, 9% Ni steel has been widely used for its high strength, economic properties, and excellent fracture toughness under cryogenic temperatures. However, the plasticity and fracture toughness of 9% Ni steel decreases with decreasing temperatures and can result in catastrophic brittle fracture. The integrity assessment of engineering structures such as the 9% Ni steel inner wall of an LNG storage tank is typically performed using fracture mechanics parameters such as the stress intensity factor K, crack tip opening displacement (CTOD), and the J integral. These conventional standard test methods have gained credibility over a long period of time by using deep notched specimens subjected to predominantly bending loads. However, cracked engineering structures are subjected to tensile loading that results in expanding plasticity and decreasing stresses near the crack tip due to the loss of plastic constraints. Therefore, using the fracture toughness values obtained from standard test specimens to represent the structural integrity under such a low constraint condition results in engineering structures that are overly conservative and thus waste time and cost [3,4,5].

Recently, increasing demand for an appropriate assessment method for the structural integrity and fitness-for-service (FFS) analysis of engineering structures has resulted in constraint effects becoming one of the more important issues in fracture mechanics and structural integrity research. Previous studies have accordingly been conducted to clarify the effect of crack tip constraints in order to ensure the appropriate assessment of engineering structures [5,6]. Two-parameter fracture mechanics (TPFM) propose that the constraint effect could be solved by introducing an additional parameter. Traditionally the most widely used additional parameters are the T-stress for the stress intensity factor K and the Q-parameter for the J integral [7,8,9], which allows investigators to perform more precise structural integrity assessments of engineering structures [10,11,12]. Advanced research into the constraint-based assessment of engineering structures has established two engineering approaches: FITNET [13,14,15] and ISO 27306 [16]. Both approaches employ the Weibull stress fracture criterion by applying the local approach, as this provides a useful theoretical foundation for a wide range of practical applications to describe brittle fractures in engineering structures. The Weibull stress σ_w is used as the fracture driving force and obeys the two-parameter Weibull distribution at the onset of brittle fracture irrespective of specimen geometry, loading type, or stress mismatching [17,18,19]. Notably, Minami [20] established a new method of assessing the effects of constraint loss on the instance of cleavage fracture by conducting CTOD tests on wide plate (WP) specimens subjected to membrane stress. The concept of the equivalent CTOD ratio β, defined as β = δ/δ_WP (where δ and δ_WP are the CTOD of a standard fracture toughness specimen and the CTOD of the WP specimen, respectively), was used to mitigate the excessive conservatism observed in the conventional procedure [20,21].

The present study accordingly focused on the effect of crack tip constraint on the fracture assessment of 9% Ni steel for application in LNG storage tanks using CTOD tests conducted using both conventional standard three-point bending (3PB) and WP specimens under a cryogenic temperature of −196 °C. The distributions of the stress and strain fields near the crack tip in the 3PB and WP specimens were then obtained through FE analysis. Based on both the theoretical and experimental results, the parameters of the Weibull distribution of the critical Weibull stress at brittle fracture were then obtained. Next, the CTOD fracture toughness value obtained from the 3PB standard specimen was converted to that of the WP standard specimen by applying an equivalent CTOD ratio β, enabling the fracture assessment of 9% Ni steel by using a constraint-based failure assessment diagram (FAD). Finally, it was demonstrated possible to perform a more carefully adjusted, less conservative assessment by incorporating the CTOD correction for constraint loss in the 9% Ni steel structure of an LNG storage tank.

## 2. Experiments

Commercial 33 mm, 37 mm, and 40 mm thick plates of 9% Ni steel were evaluated in this study. Table 1 and Table 2 show their chemical composition and mechanical properties. Figure 1 and Figure 2 show the configuration of a 3PB and WP CTOD specimen, respectively, containing a center through-thickness crack. The 3PB CTOD specimens were prepared in 29 mm, 33 mm, and 36 mm thicknesses after a 4-mm reduction for surface treatment. As shown in Figure 1, the width W, length L, and notch shape were determined in terms of the specimen thickness B. The thicknesses of the WP CTOD specimens were prepared at the original thicknesses of the fabricated plate. Fatigue pre-cracks were introduced at the tip of the machined notch in all the specimens before performing the fracture test. The 3PB CTOD fracture toughness tests were performed in compliance with BS 7448 Part I [22] using 50 ton dynamic universal testing machine and those for the WP specimens were performed using 3000- on a special testing machine. To determine the CTOD of the WP specimens, the crack opening displacements were measured at the near end of the notch by clip gages installed as shown in Figure 2.

The CTOD values of the WP specimens were calculated using the BCS model [20,23]. All the fracture tests were conducted at −196 °C. The CTOD values of the 3PB and WP specimens are shown in Figure 3. Thickness clearly has no effect on the critical CTOD values for either the 3PB or the WP specimens, so the thickness effect is not considered in this study. Figure 3 shows that the WP specimens have a higher CTOD fracture toughness than the 3PB specimens because the tensile loading mode of the WP specimens results in a smaller constraint on the crack tip compared to the bending loading mode of the 3PB specimens. The constraint loss effect should clearly be taken into account when assessing a 9% Ni steel engineering structure.

## 3. Evaluation of Fracture Toughness Using the Local Approach

### FE Analysis of Near Crack-Tip Fields

Near crack-tip stress fields play an important role in determining the CTOD value [20,24]. To evaluate the effects of constraint loss on the crack tip stress fields, the CTOD specimens were analyzed using a 3D finite element model in ABAQUS (Ver. 6.14-5, Dassault Systèmes Simulia Corp. Providence, RI, USA). As the specimens were biaxially symmetric, only one-quarter of each CTOD specimen was modeled. The FE-models used in this study are shown in Figure 4 and Figure 5. The 8-node element was used for the specimens and the minimum size of this element was set to 0.003 mm × 0.003 mm × 0.006 mm. Details of the refined mesh near the crack tip stress fields are also shown in Figure 4 and Figure 5.

The crack tip stress fields were analyzed by increasing the CTOD value. The distributions of the crack opening stress $\sigma_{\mathrm{yy}}$ near the crack tip in the middle of the 3PB and WP CTOD specimen thicknesses are shown in Figure 6 and Figure 7, respectively, for different CTOD values. In Figure 6, it can be observed that even with increased CTOD values, the highest opening stress and crack tip positions were largely maintained in the 3PB specimens; this is likely due to the higher constraint on the crack tip of 3PB specimens under the bending load. The maximum opening stress for the WP specimens with a CTOD of 0.26 was similar to that of the 3PB specimens with a CTOD of 0.28. However, though the maximum opening stress of the WP specimens with a CTOD of 0.45, 0.78, and 0.83 remain the same, the opening stresses at locations far from the crack tip decrease with the increasing CTOD value due to the smaller constraint was applied on the crack tip of the WP specimens under tensile loading.

Figure 8 shows the appearance of the specimens’ fracture after the 3PB specimen CTOD test. The progression of the front line of the crack is similar throughout, from the middle of the crack to the specimen surface. At the surface area of the specimen, a small shear lip formed, which exhibits a traditional crack propagation shape under bending load. Figure 9 and Figure 10 show the FE analysis results of the stress distribution along the specimen thickness similar to the test results. In contrast, the wide specimen shows different results, as shown in Figure 11. The progression of the front line of the crack is large at the middle of the specimen thickness. At the surface area, a relatively large shear lip is formed, and it exhibits a traditional crack propagation shape under a tensile mode. Figure 12 and Figure 13 show the FE analysis results of the stress distribution along the specimen thickness. At the low CTOD levels of 0.26 and 0.45, the cracks show a similar stress distribution with the 3PB results; however, at the high CTOD levels of 0.75 and 0.83, the cracks show an opening stress value that sharply decreased as they approached the surface of the specimen. It is noted that the stress distribution along the thickness is highly dependent on the stress mode and CTOD levels. Lower stress level distribution at the surface of the wide specimen inhibits the brittle fracture.

## 4. Weibull Stress Criterion

The Weibull stress criterion was established for the assessment of the brittle fracture of metallic ferritic materials using the local approach. The local approach employs the Weibull stress *σ_w_* as the fracture driving force of the materials. This paper employs the Weibull stress criterion for the evaluation of the stress distribution and fracture behavior of the 3PB and WP CTOD specimens. The Weibull stress is given by the integration of the near crack-tip stresses in the fracture process zone as follows:(1)$$\sigma_{w}={\left[\frac{1}{V_{0}}\int_{V_{P}}^{}\sigma_{1}{}^{m}dV_{P}\right]}^{1/m}$$
where $V_{P}$ is the volume of the fracture process zone, $V_{0}$ is a reference volume, m is the shape parameter, and $\sigma_{1}$ is the maximum principal stress. The critical Weibull stress $\sigma_{w,cr}$ at the initiation of the brittle fracture obeys the Weibull distribution with two parameters m and $\sigma_{u}$, where both parameters are assumed to be material properties independent of the loading rate and specimen geometry. The m value implies the distribution of microcracks in the material. Figure 8 indicates that the 3PB and WP specimens exhibit similar critical Weibull stress distributions. The Weibull stress distribution of the 3PB fracture toughness data is approximated by the line in Figure 14. The shape parameter of this distribution is m = 21.9 and the scale parameter is $\sigma_{u}$ = 2.318 MPa.

## 5. Failure Assessment and Constraint Loss Correction Using Equivalent CTOD Ratio

Correcting for constraint loss is of critical importance to an accurate evaluation of structural integrity. As shown in Figure 3, the fracture toughness data obtained from the 3PB specimens were considerably smaller than those obtained from the WP specimens. Applying the 3PB specimen CTOD data to evaluate an engineering structure under tensile loading conditions clearly provides a very conservative result. Generally, the fracture assessment of an engineering structure is conducted using the FAD as specified in BS7910 [25]. FAD consists of two axes: the linear elastic stress intensity factor $L_{r}$, normalized by the material fracture toughness on the horizontal axis and the toughness ratio $\sqrt{\delta_{r}}$ on the vertical axis. The failure assessment curve (FAC) for the evaluation of the safety at level 2 of the FAD is calculated using
(2)$$\sqrt{\delta_{r}}=f\left(L_{r}\right)=\sqrt{\frac{\delta_{\mathrm{WP}}^{e}}{\delta_{\mathrm{mat}}}}=\left(1-0.14L_{r}^{2}\right)\left[0.3+0.7\exp\left(-0.65L_{r}^{6}\right)\right]$$
where $\sqrt{\delta_{r}}$ is the toughness ratio, $\delta_{\mathrm{WP}}^{e}$ is the elastic CTOD of the WP specimen, $\delta_{\mathrm{mat}}$ is the critical CTOD of the material, and $L_{r}$ is the load ratio (=$\sigma_{\mathrm{ref}}/\sigma_{y}$), in which $\sigma_{\mathrm{ref}}$ and $\sigma_{y}$ are, respectively, the reference stress representing the average stress at the net cross section of the cracked WP specimen and the yield stress at −196 °C. The elastic component of the WP CTOD, $\delta_{\mathrm{WP}}^{e}$, is calculated from the stress intensity factor K by
(3)$$\delta_{\mathrm{WP}}^{e}=\frac{K^{2}}{E^{\prime }\sigma_{y}}$$
where $\sigma_{y}$ is the yield stress at −196 °C, $E^{\prime }=E$ when representing the plane stress, and $E^{\prime }=E\left(1-\nu^{2}\right)$ when representing the plane strain, in which E is the Young’s modulus of the specimen.

The proposed procedure converts the 3PB specimen CTOD fracture toughness to the WP specimen CTOD according to their correlations to the Weibull stress $\sigma_{w}$ as per the assessment of structural components using FAD-based methods [20,21,26]. The precision of the proposed procedure can be ensured by applying the equivalent CTOD ratio $\beta =\delta /\delta_{\mathrm{WP}}$, shown in Figure 15, where δ and $\delta_{\mathrm{WP}}$ are the CTOD of a conventional 3PB fracture toughness specimen and the CTOD of a WP specimen, respectively. The correlation of the experimental CTOD values with the theoretical Weibull stress values for the 3PB and WP specimens were analyzed and plotted on the FAD as shown in Figure 16 when the fracture toughness is corrected using the equivalent CTOD ratio β. In this figure, the ■ symbol indicates the method of directly applying $\delta_{\mathrm{mat}}$ as the critical CTOD value $\delta_{\mathrm{cr}}$ of the 3PB standard fracture toughness results, whereas the □ symbol indicates that the fracture toughness values have been corrected by taking into account the equivalent CTOD ratio β using $\delta_{\mathrm{mat}}=\delta_{\mathrm{cr}}/\beta$.

When using the conventional evaluation procedure with the CTOD values of the 3PB specimens, the resulting fracture ratios lie in an unsafe area above the curve in Figure 10. However, when applying the CTOD correction ratio β, the resulting fracture ratios are reasonably reduced to within the safe area below the curve. This reduction reflects the difference in the constraint conditions at the crack tip area under the bending loading of the 3PB specimen and the tensile loading of the WP specimen. Thus, it has been demonstrated that when applied to materials in tension, the conventional CTOD evaluation method results in an excessively conservative result that can be reasonably reduced by applying the equivalent CTOD ratio β.

## 6. Conclusions

In this study, an appropriate assessment method for the fracture toughness of a 9% Ni steel for use in an LNG storage tank was proposed and conducted under a cryogenic temperature considering the difference in testing constraint effects on the material CTOD. The utility of this method was demonstrated using the fracture toughness test results, FE modeling, Weibull stress criterion, and FAD analysis. The following conclusions were obtained from this study:(1)WP specimens exhibited a larger CTOD fracture toughness than 3PB specimens. This is due to the fact that the tensile loading mode exerts a smaller constraint effect on the crack tip of the WP specimens than the bending loading mode applied to the 3PB specimens.(2)It was proven that the Weibull stress fracture criterion is a useful fracture parameter for the assessment of brittle fracture of 9% Ni steel when using the local approach. The Weibull stress $\sigma_{w}$
obeys the two-parameter Weibull distribution at the onset of brittle fracture in both the 3PB and WP specimens.(3)The conventional fracture toughness evaluation procedure resulted in an excessively conservative result that can be reasonably reduced by applying the equivalent CTOD ratio β.
This ratio can be obtained by analyzing the correlation between experimental CTOD values and the theoretical Weibull stress for the 3PB and WP specimens.

## Figures and Tables

**Figure 1 materials-13-05250-f001:**
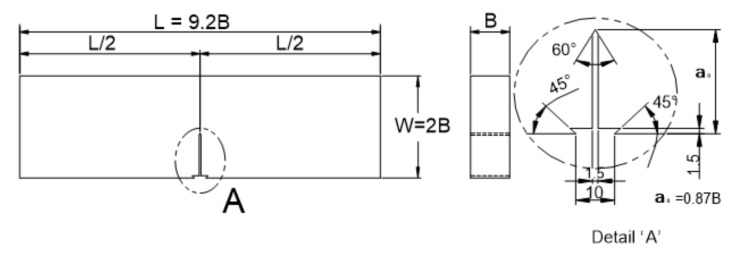
CTOD 3PB specimen (unit: mm).

**Figure 2 materials-13-05250-f002:**
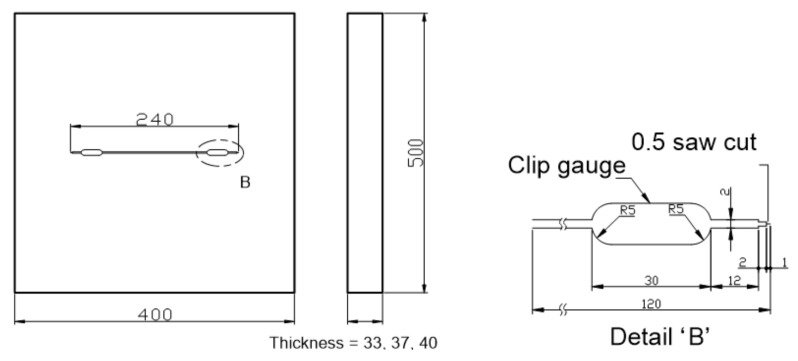
CTOD WP specimen (unit: mm).

**Figure 3 materials-13-05250-f003:**
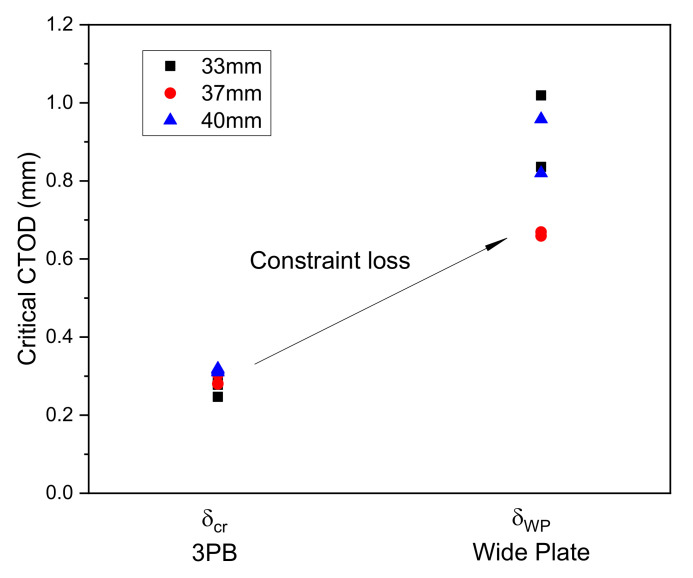
Comparison of CTOD values determined using 3PB and WP tests.

**Figure 4 materials-13-05250-f004:**
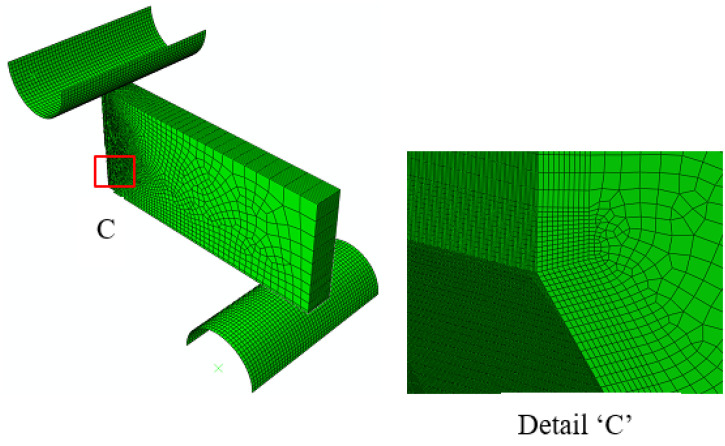
Finite element model of the 3PB bending specimen.

**Figure 5 materials-13-05250-f005:**
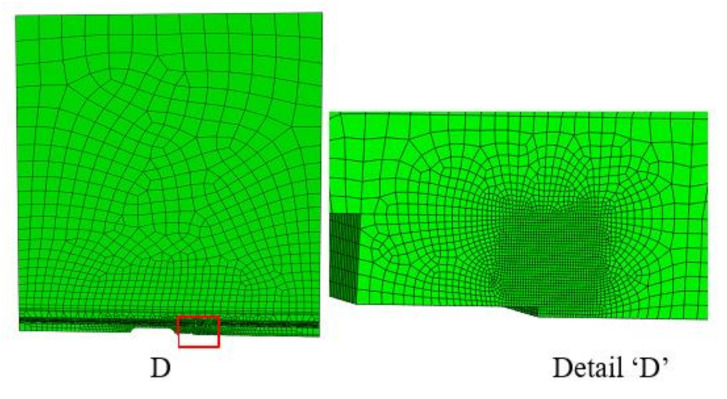
Finite element model of the WP specimen.

**Figure 6 materials-13-05250-f006:**
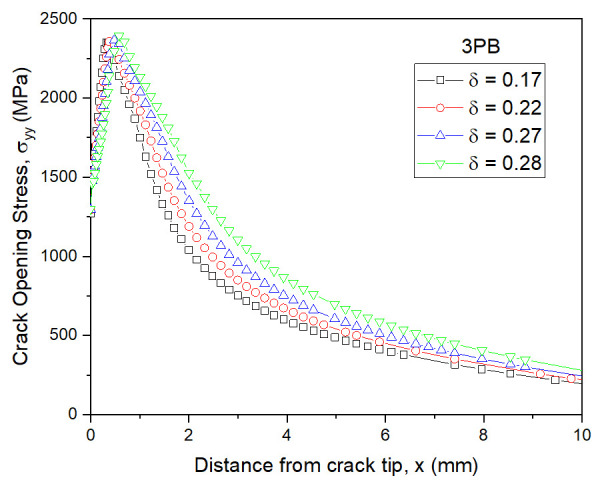
Opening stress distributions of the 3PB specimens under different CTODs.

**Figure 7 materials-13-05250-f007:**
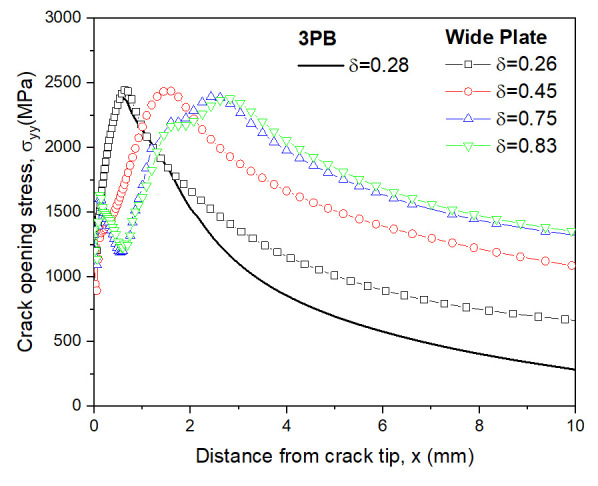
Opening stress distributions of WP and 3PB specimens under different CTODs.

**Figure 8 materials-13-05250-f008:**
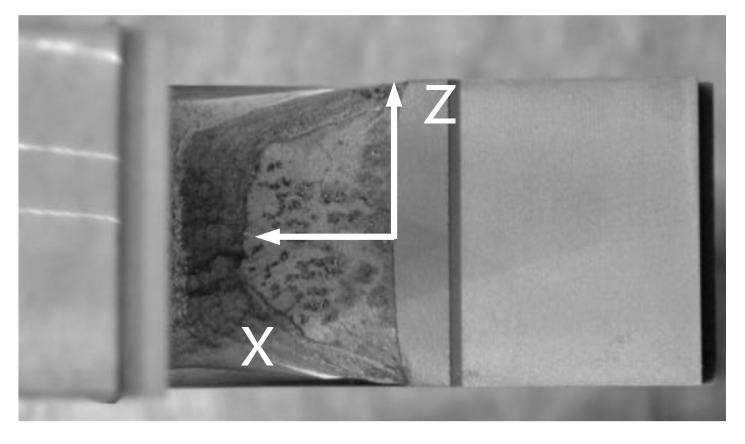
Appearance of fracture in a 3PB CTOD test specimen (33 mm thickness).

**Figure 9 materials-13-05250-f009:**
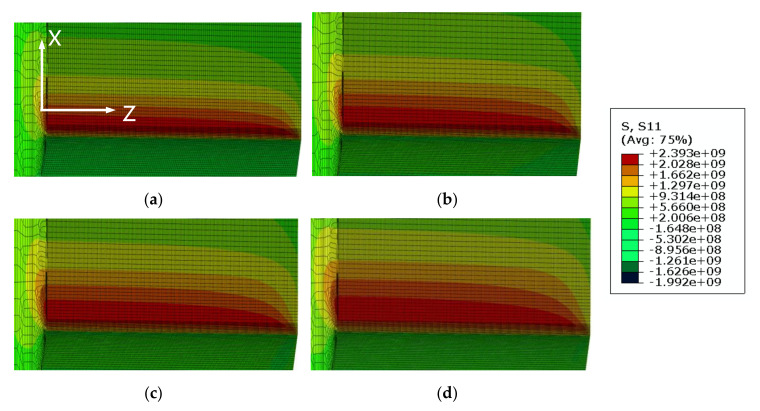
Opening stress distributions from FE analysis of 3PB specimens (**a**) CTOD 0.17 mm (**b**) CTOD 0.22 mm (**c**) CTOD 0.27 mm (**d**) CTOD 0.28 mm.

**Figure 10 materials-13-05250-f010:**
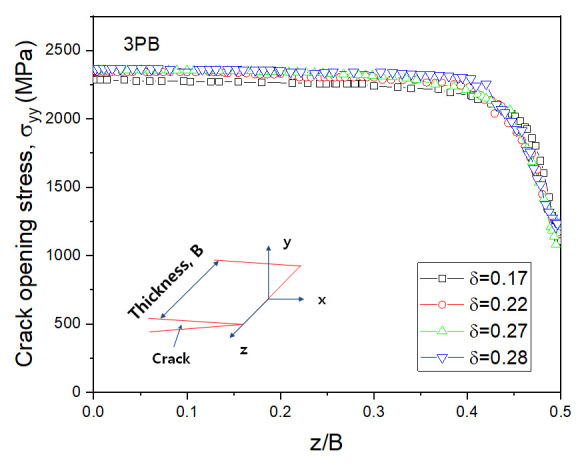
Variation of opening stress with the thickness for 3PB specimen.

**Figure 11 materials-13-05250-f011:**
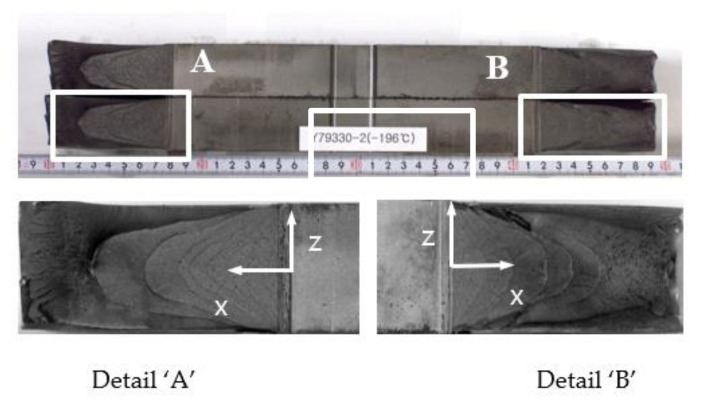
Appearance of fracture in wide plate test specimen (33 mm thickness).

**Figure 12 materials-13-05250-f012:**
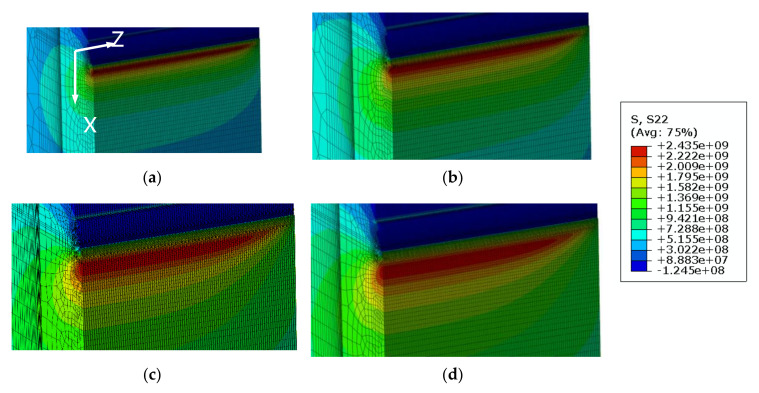
Opening stress distributions from FE analysis of wide plate specimens (**a**) CTOD 0.26 mm (**b**) CTOD 0.45 mm (**c**) CTOD 0.75 mm (**d**) CTOD 0.83 mm.

**Figure 13 materials-13-05250-f013:**
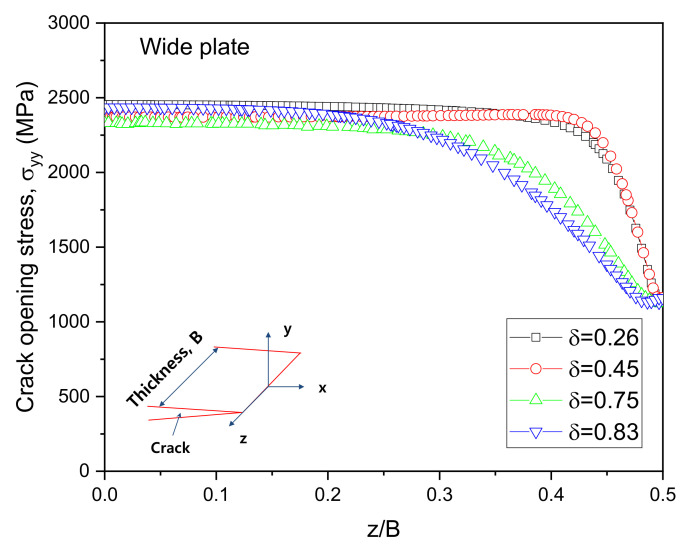
Opening stress distribution along the thickness for wide plate specimen.

**Figure 14 materials-13-05250-f014:**
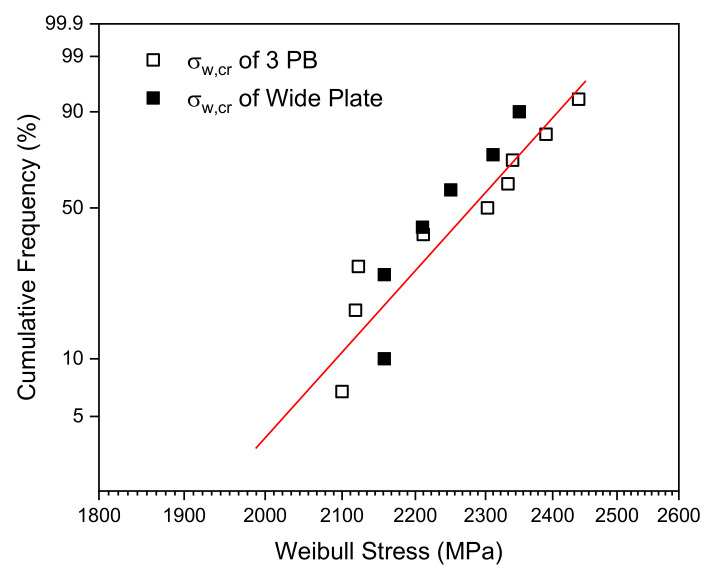
Critical Weibull stress distributions for 3PB and WP specimen fracture toughness test data.

**Figure 15 materials-13-05250-f015:**
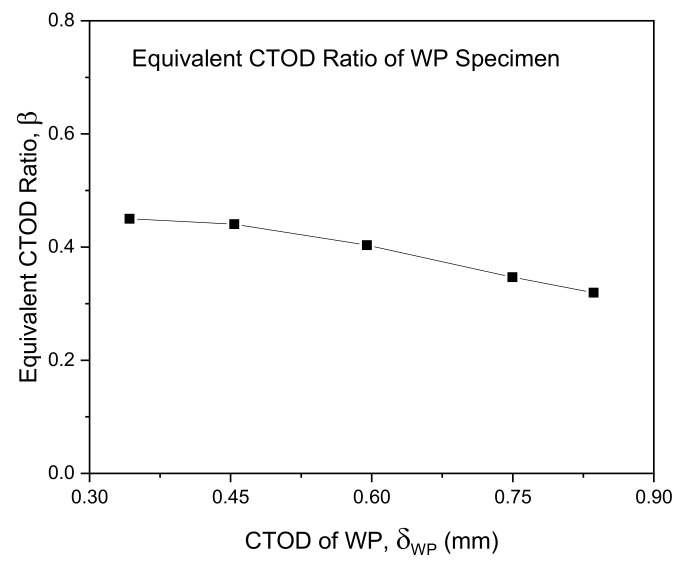
Equivalent CTOD ratio of the WP specimens.

**Figure 16 materials-13-05250-f016:**
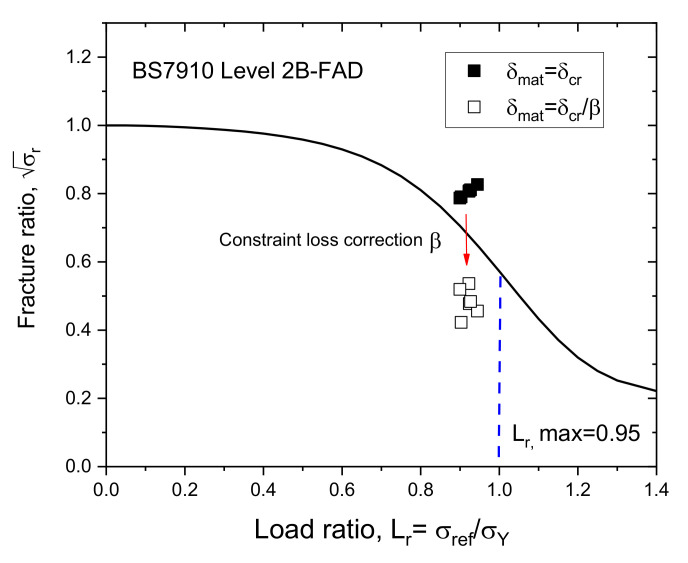
FAD estimation of 9% Ni steel with and without accounting for the equivalent CTOD ratio.

**Table 1 materials-13-05250-t001:** Chemical composition of 9% Ni steel in wt. % with Fe to balance.

	C	Mn	P	S	Si	Ni
Chemical Composition (wt.%)	0.04	0.61	0.07	0.005	0.003	9.04

**Table 2 materials-13-05250-t002:** Mechanical properties of 9% Ni steel at −196 °C.

Yield Stress $\sigma_{Y_{0}}$ (MPa)	Tensile Strength $\sigma_{T}$ (MPa)	Elongation $\varepsilon_{f}$ (%)
941	1091	30.3

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
