# Peer review of "Effects of Crack Tip Constraint on the Fracture Toughness Assessment of 9% Ni Steel for Cryogenic Application in Liquefied Natural Gas Storage Tanks"

_materials, 2020, doi:10.3390/ma13225250_

Round 1
Reviewer 1 Report
The study provides a better method for analyzing crack initialing stress and method of estimating the fracture toughness precisely. The presented FEM model also provides enough evidence to explain the experimental results. Its quality is enough to accept publication.
Author Response
Point 1 : The study provides a better method for analyzing crack initialing stress and method of estimating the fracture toughness precisely. The presented FEM model also provides enough evidence to explain the experimental results. Its quality is enough to accept publication.
Response 1 :
Reviewer 2 Report
The paper is well written and explained. To pinpoint some possible improvements:
- I am missing some scale bars in the optical images provided in Fig. 8 and 11.
- The authors should separate 9 and %.
- In the Figure caption of Fig. 2, the authors should mention that the values are provided in mm. In this figure, 0.5 and mm have to be separated
- In the title of Table 1 it shoudl be written "with Fe to balance"
- Mechanical properties in Table 2 should have an experimental error. Were these properties determined by the authors or obtained from other sources? Experimental values should always include and experimental error.
- Figure caption of Fig. 8: separate 33 and mm.
Author Response
Point 1: I am missing some scale bars in the optical images provided in Fig. 8 and 11.
Response 1: The thickness of specimen in Fig. 8 and 11 are 33 mm. We thought there is no need of scale bar displaying in the figures
Point 2: The authors should separate 9 and %.
Response 2: Generally using the steel name in the phrase of “9% Ni” as shown in the papers.
(1) Effect of dispersed retained γ-Fe on brittle crack arrest toughness in 9% Ni steel in cryogenic temperatures
- https://doi.org/10.1016/j.msea.2018.03.056
(2) Fracture toughness of a 9% Ni steel pipe girth welded with Ni-based superalloy 625 filler metal operating in a sour environment
- https://doi.org/10.1016/j.jmrt.2020.03.044
Point 3: In the Figure caption of Fig. 2, the authors should mention that the values are provided in mm. In this figure, 0.5 and mm have to be separated
Response 3: Our manuscript will be revised according to your comments.
Point 4: In the title of Table 1 it shoudl be written "with Fe to balance"
Response 4: Our manuscript will be revised according to your comments.
Point 5: Mechanical properties in Table 2 should have an experimental error. Were these properties determined by the authors or obtained from other sources? Experimental values should always include and experimental error.
Response 5: We have obtained the mechanical properties by performing experiment as shown in the below table. In this paper we have focused on the fracture behavior of 3PB and WP specimens so we simply described the mechanical properties as shown in Table 2.
|
Specimen No. |
Yield Strength (MPa) |
Tensile Strength (MPa) |
Elongation (%) |
|
|
33mm Longitudinal |
1 |
938.5 |
966.7 |
33.5 |
|
2 |
950.3 |
993.2 |
31.4 |
|
|
3 |
944.4 |
980 |
31.2 |
|
|
Average |
944.4 |
980.0 |
32.0 |
|
|
33mm Transverse |
1 |
938.5 |
975.6 |
31.3 |
|
2 |
941.9 |
1065.0 |
29.0 |
|
|
3 |
936.2 |
1069.0 |
29.6 |
|
|
Average |
938.9 |
1,036.5 |
30.0 |
|
|
37mm Longitudinal |
1 |
900.8 |
1101.4 |
29.8 |
|
2 |
892.4 |
1091.2 |
30.9 |
|
|
3 |
883.9 |
1081.1 |
30.2 |
|
|
Average |
892.4 |
1,091.2 |
30.3 |
|
|
37mm Transverse |
1 |
916.1 |
1092.8 |
27.8 |
|
2 |
789.3 |
916.8 |
29.1 |
|
|
3 |
943.3 |
1096.0 |
30.2 |
|
|
Average |
882.9 |
1,035.2 |
29.0 |
|
|
40mm Longitudinal |
1 |
939.1 |
1110.5 |
32.4 |
|
2 |
938.2 |
1112.6 |
34.0 |
|
|
3 |
948.3 |
1112.1 |
33.1 |
|
|
Average |
941.9 |
1,111.7 |
33.2 |
|
|
40mm Transverse |
1 |
944.4 |
1100.1 |
32.2 |
|
2 |
947.6 |
1108.6 |
31.5 |
|
|
3 |
941.2 |
1099.6 |
29.4 |
|
|
Average |
944.4 |
1,102.8 |
31.0 |
|
Point 6: Figure caption of Fig. 8: separate 33 and mm.
Response 6: Our manuscript will be revised according to your comments.
Reviewer 3 Report
The intention of the authors of the article was to propose a suitable method for assessing the fracture toughness of 9% Ni steel used as a construction material for liquefied natural gas tanks. Due to LNG storage conditions, the experiments were performed at a cryogenic temperature of -196 ° C.
The performed experimental research described in the article uses standard methods for assessing the fracture toughness of materials. They combine experimental research with theoretical FE modeling and the use of the Weibull stress criterion and FAD analysis.
I consider the authors' approach to be scientific and the results of the work are interesting.
Comments:
- The article lacks a more comprehensive comparison of the proposed method and conclusions of experiments with similar work of other research teams.
- Lines 100-101 are missing the description of Figure 1.
Author Response
Point 1: The article lacks a more comprehensive comparison of the proposed method and conclusions of experiments with similar work of other research teams.
Response 1: We have investigated the 9% Ni material fracture toughness behavior by performing experiment and FE analysis under tensile and bending load conditions.
Point 2: Lines 100-101 are missing the description of Figure 1.
Response 2: Our manuscript will be revised according to your comments,
Reviewer 4 Report
18: The full name for FE (Finite Elements?) is missing.
63: The reference [20] should come directly after the name "Minami"
83 and 86: Figure 1 is mentioned but does not exist.
93, 94 and 100: If the gauges exist at both ends of the crack, a plural is to be used in line 94.
107: It is noted in the publication that the thickness of the sample does not play a role, but in FIG. 3
the critical CTOD are between 0.6 and 1 mm. Please provide a more detailed explanation.
165: The picture could be larger and the shear lips can be labeled in the picture.
166 and 173: The units in Figures 9 and 12 are missing.
180: σ_w should be σw.
176: Figure 13 is not mentioned in the text.
190: Is Figure 8 really meant here?
Author Response
Review 1: 18: The full name for FE (Finite Elements?) is missing.
Response 1: Our manuscript will be revised according to your comments.
Review 2: 63: The reference [20] should come directly after the name "Minami"
Response 2: Our manuscript will be revised according to your comments.
Review 3: 83 and 86: Figure 1 is mentioned but does not exist.
Response 3: It was our mistake. Our manuscript will be revised according to your comments.
Review 4: 93, 94 and 100: If the gauges exist at both ends of the crack, a plural is to be used in line 94.
Response 4: Our manuscript will be revised according to your comments.
Review 5: 107: It is noted in the publication that the thickness of the sample does not play a role, but in FIG. 3 the critical CTOD are between 0.6 and 1 mm. Please provide a more detailed explanation.
Response 5: According to many references reported the similar variation of CTOD values under the condition of -196℃. Especially there is larger variation of CTOD values under room temperature. So we made a conclusion three was no effect of thickness on the specimen.
Review 6: 165: The picture could be larger and the shear lips can be labeled in the picture.
Response 6: The purpose of Fig. 8 shows the crack front line is similar with Fig. 10 under the condition of bending load condition.
Review 7: 166 and 173: The units in Figures 9 and 12 are missing.
Response 7: Our manuscript will be revised according to your comments.
Review 8: 180: σ_w should be σw.
Response 8: Our manuscript will be revised according to your comments.
Review 9: 176: Figure 13 is not mentioned in the text.
Response 9: Figure 13 is mentioned in the line of 150-154. We have compared Figure 8 and Figure 13 to investigate the distribution of stress under the condition of bending and tensile loading conditions.
Review 10: 190: Is Figure 8 really meant here?
Response 10: Please refer to Response 6 and 9.
Reviewer 5 Report
The paper details an investigation which, despite not being highly innovative in terms of the topic, shows some interesting points.
There are however some aspects which need to be clarified.
In details:
- The authors state the used material is commercial. The producer should be specified. Moreover the chemical should be provided with tolerances applied to the elemental % (tab 1)
- Where do the values reported in Tab 2 come from? literature analysis or datasheet of the material?
- Which type of testing machines have been used? (producer and model)
- Which were the conditions of testing?
- How many samples have been tested and which are the Dev St. obtained?
- Line 105/106: please provide deeper description on how tests at -196°C have been performed
- Fig 4 and 5 appear to be "oversized" and may eventually be merged
- Fig 8 does not show anything. A much higher magnification should be provided, as well as a discussion of the (microstructural) areas where the fracture propagate
- Same concepts apply to fig 11
Author Response
Point 1: The authors state the used material is commercial. The producer should be specified. Moreover the chemical should be provided with tolerances applied to the elemental % (tab 1)
Response 1: We will apply the producer in the paper.
The detail data of chemical compositions is shown as below. In this paper we have focused on the fracture behavior of 3PB and WP specimens so we simply described the chemical composition as shown in Table 2.
|
Classification |
Chemical Composition (wt%) |
||||||
|
C |
Mn |
Si |
P |
S |
Ni |
||
|
33mm |
1/2t |
0.05 |
0.70 |
0.25 |
0.006 |
0.003 |
8.95 |
|
1/4t |
0.05 |
0.69 |
0.25 |
0.006 |
0.003 |
8.90 |
|
|
37mm |
1/2t |
0.05 |
0.69 |
0.24 |
0.007 |
0.003 |
8.84 |
|
1/4t |
0.05 |
0.69 |
0.24 |
0.006 |
0.003 |
8.79 |
|
|
40mm |
1/2t |
0.06 |
0.70 |
0.26 |
0.004 |
0.003 |
8.90 |
|
1/4t |
0.06 |
0.70 |
0.26 |
0.004 |
0.003 |
8.89 |
|
Point 2: Where do the values reported in Tab 2 come from? literature analysis or datasheet of the material?
Response 2: We have obtained the mechanical properties by performing experiment as shown in the below table. In this paper we have focused on the fracture behavior of 3PB and WP specimens so we simply described as shown in Table 2.
|
Specimen No. |
Yield Strength (MPa) |
Tensile Strength (MPa) |
Elongation (%) |
|
|
33mm Longitudinal |
1 |
938.5 |
966.7 |
33.5 |
|
2 |
950.3 |
993.2 |
31.4 |
|
|
3 |
944.4 |
980 |
31.2 |
|
|
Average |
944.4 |
980.0 |
32.0 |
|
|
33mm Transverse |
1 |
938.5 |
975.6 |
31.3 |
|
2 |
941.9 |
1065.0 |
29.0 |
|
|
3 |
936.2 |
1069.0 |
29.6 |
|
|
Average |
938.9 |
1,036.5 |
30.0 |
|
|
37mm Longitudinal |
1 |
900.8 |
1101.4 |
29.8 |
|
2 |
892.4 |
1091.2 |
30.9 |
|
|
3 |
883.9 |
1081.1 |
30.2 |
|
|
Average |
892.4 |
1,091.2 |
30.3 |
|
|
37mm Transverse |
1 |
916.1 |
1092.8 |
27.8 |
|
2 |
789.3 |
916.8 |
29.1 |
|
|
3 |
943.3 |
1096.0 |
30.2 |
|
|
Average |
882.9 |
1,035.2 |
29.0 |
|
|
40mm Longitudinal |
1 |
939.1 |
1110.5 |
32.4 |
|
2 |
938.2 |
1112.6 |
34.0 |
|
|
3 |
948.3 |
1112.1 |
33.1 |
|
|
Average |
941.9 |
1,111.7 |
33.2 |
|
|
40mm Transverse |
1 |
944.4 |
1100.1 |
32.2 |
|
2 |
947.6 |
1108.6 |
31.5 |
|
|
3 |
941.2 |
1099.6 |
29.4 |
|
|
Average |
944.4 |
1,102.8 |
31.0 |
|
Point 3: Which type of testing machines have been used? (producer and model)
Response 3: We have performed 3PB bending system by using Instron 8504. And for performing the wide plate specimen by using specially designed test machine of Shimadzu company.
Point 4: Which were the conditions of testing?
Response 4: All the test discussed in the paper were performed under the condition of -196℃.
Point 5: How many samples have been tested and which are the Dev St. obtained?
Response 5: 6 specimens have been tested for 3PB bending and wide plate specimen respectively.
Review 5: Line 105/106: please provide deeper description on how tests at -196°C have been performed
Response 5: Temperature condition has been obtained using a bath with liquid nitrogen.
Review 6: Fig 4 and 5 appear to be "oversized" and may eventually be merged
Response 6: Different type of specimens FE element might be displayed respectively as shown in Fig. 1~2.
Review 7: Fig 8 does not show anything. A much higher magnification should be provided, as well as a discussion of the (microstructural) areas where the fracture propagate Same concepts apply to fig 11
Response 7: We have compared Figure 8 and Figure 13 to investigate the distribution of stress under the condition of bending and tensile loading conditions. And also compare the crack front line with the FE analysis results as shown in Figure 9 and Figure 12 respectively.
Round 2
Reviewer 5 Report
The authors have answered to all the questions and made modifications of the paper in a satisfactory way. The article may therefore go for publication